# Adaptive mask-based brain extraction method for head CT images

**Dingyuan Hu[1]☯, Shiya Qu[1]☯, Yuhang Jiang[1], Chunyu Han[1], Hongbin Liang☯[1]***, Qingyan Zhang[2]**

**1** School of Mechanical Engineering and Automation, University of Science and Technology Liaoning, Qianshan District, Anshan City, Liaoning Province, China, **2** Radiology, Ninth People's Hospital of Zhengzhou, Jinshui District, Zhengzhou City, Henan Province, China

☯ These authors contributed equally to this work.
* lianghb@hit.edu.cn

**Data Availability Statement:** The datasets were derived from the RSNA Intracranial Hemorrhage Original Size PNGs (RIHOSP)dataset, publicly available on the Kaggle website, and the CQ500

## Abstract

Brain extraction is an important prerequisite for the automated diagnosis of intracranial lesions and determines, to a certain extent, the accuracy of subsequent lesion identification, localization, and segmentation. To address the problem that the current traditional image segmentation methods are fast in extraction but poor in robustness, while the Full Convolutional Neural Network (FCN) is robust and accurate but relatively slow in extraction, this paper proposes an adaptive mask-based brain extraction method, namely AMBBEM, to achieve brain extraction better. The method first uses threshold segmentation, median filtering, and closed operations for segmentation, generates a mask for the first time, then combines the ResNet50 model, region growing algorithm, and image properties analysis to further segment the mask, and finally complete brain extraction by multiplying the original image and the mask. The algorithm was tested on 22 test sets containing different lesions, and the results showed MPA = 0.9963, MIoU = 0.9924, and MBF = 0.9914, which were equivalent to the extraction effect of the Deeplabv3+ model. However, the method can complete brain extraction of approximately 6.16 head CT images in 1 second, much faster than Deeplabv3+, U-net, and SegNet models. In summary, this method can achieve accurate brain extraction from head CT images more quickly, creating good conditions for subsequent brain volume measurement and feature extraction of intracranial lesions.

## 1. Introduction

As brain morbidity continues to increase, computed tomography (CT) and magnetic resonance imaging (MRI) play a vital role in diagnosing intracranial lesions as brain imaging modalities [1]. Compared to MRI, CT is widely used because of its lower cost and faster diagnosis. In recent years, in order to assist radiologists in making accurate and rapid diagnoses, domestic and foreign researchers have been dedicated to applying image processing techniques to medical image analysis [2] to achieve segmentation or detection of brain tumors [3], intracranial hematoma [4, 5]. . . and other lesions. However, for diagnosing intracranial lesions, not all of the information provided by CT scans is useful. For example, diagnostic equipment, pillows, skulls, and other non-brain tissues are not useful and can largely affect the

dataset on the Academic Torrents website. No patient privacy was involved. The training and test sets mentioned in the paper are obtained from the (RIHOSP) dataset and CQ500 dataset after screening and processing. We have put the training and test sets mentioned in the experiments in the Supporting information, which can be used directly. As for the (RIHOSP) dataset and CQ500 dataset, they are also available at the following links. The link to the (RIHOSP) dataset is https://www.kaggle.com/datasets/vaillant/rsna-ich-png/. Download Methods: Click the link to enter the webpage, then click "Download" to download it directly. It should be noted that the download requires an account to log in. The account we provide is account a18737263685; password: hu123456. The link to the CQ500 data collection is https://academictorrents.com/details/47e9d8aab761e75fd0a81982fa62bddf3a173831. Download Methods: First, click the link to enter the webpage; then click "Download" to download a TORRENT file (named "qure.headct.study-47e9d8aab761e75fd0a81982fa62bddf3a173831.torrent"). Finally, upload the TORRENT file to the download utility for direct download.

**Funding:** This research was supported by the Department of Science and Technology of Liaoning Province, Natural Foundation Project (No: 2015020128). The funders had no role in study design, data collection and analysis, the decision to publish, or the preparation of the manuscript.

**Competing interests:** The authors have declared that no competing interests exist.

algorithm's ability to identify, localize, and segment intracranial lesions. Extracting the brain from CT images can provide a better environment for subsequent feature extraction of intracranial lesions [6], which can improve the accuracy of subsequent lesion detection, so extracting the brain from CT images has significant research significance. However, CT images of the head are complex. From a tomographic anatomical point of view, the human head can be divided from bottom to top into the basis cranii layer, the sella turcica layer, the suprasellar cistern layer, the third ventricle layer, the third ventricle top layer, the lateral ventricle layer, the lateral ventricle top layer, the centrum semiovale layer, and the cerebral cortex layer, each of which is different from the other. At the same time, situations such as the unclosed skull, the distribution of multiple brain regions, and various lesions make high-quality brain extraction difficult [7].

In recent decades, researchers at home and abroad have researched brain extraction and proposed representative algorithms, roughly divided into traditional image segmentation methods, secondary development of medical image post-processing software, and deep learning models. Traditional image segmentation methods achieve the segmentation of target regions by artificially set rules. MM Kyaw et al. [8] used a tracking algorithm to perform brain parenchyma extraction, but extracranial soft tissue could not be eliminated. B. Shahangian et al. [9] used threshold segmentation, median filtering, and image and mask multiplication for brain extraction. They later built on this to achieve further segmentation of cerebral hematomas with high accuracy. N Farzaneh et al. [10] used a custom distance regularized level set evolution (DRLSE) for brain extraction before further implementing subdural hematoma segmentation.

Gautam et al. [11] and G Cao's team [12] clustered images using White Matter Fuzzy C-means (WMFCM) and Fuzzy C-means (FCM), respectively, and then used morphological imaging to extract brain parenchyma. However, none of the above algorithms was tested on the whole group of head CT images containing different lesions, and whether they can overcome large areas of soft tissue edema, unclosed skull, and multi-regional distribution of the brain remains to be examined.

The secondary development of medical image post-processing software is widely used in MR and CT. Muschelli et al. [13] modified the fractional intensity (FI) parameters. They adjusted the brain parenchyma threshold range based on the Brain Extraction Tool (BET) to achieve high-accuracy brain extraction of MR and CT images. Bauer et al. [14] developed a brain extraction method based on Insight Toolkit (ITK), which first forms a rough mask based on the original image and later uses a level set algorithm to increase the accuracy further. These methods are publicly available, but incorporating them into other systems will be challenging.

Recently, deep learning has also been widely used in brain neuroimage. DHM Nguyen et al. [15] combined the active shape model and convolutional neural network (CNN) to give full play to the advantages of two to extract the brain from head images with good results. Zeynettin Akkus et al. [16]. proposed a full convolutional neural network (FCN) based approach and tested five models, including 2D U-Net, two modified versions of 2D U-Net, 3D U-Net, and SegNet. The experimental results show that the best model has strong robustness and high accuracy, which proves the feasibility of FCN to achieve CT image brain extraction. However, the effects of different lesions on the FCN segmentation effect still need to be thoroughly tested. Comprehensive analysis shows that for brain extraction of head CT images, the FCN model has better robustness and higher accuracy compared with traditional image segmentation algorithms, but the number of parameters is vast, and the segmentation speed is slow [17, 18], and further improvements in segmentation speed while ensuring accuracy and robustness have yet to be thoroughly tried.

Here, to solve the above problems and thus better achieve brain extraction, this paper combines traditional image processing methods and CNN and proposes an integrated method, namely AMBBEM. The AMBBEM and three FCN models were tested on 22 sets of head CT images containing different lesions. The contributions of this paper are as follows: First, an integrated algorithm for brain extraction from head CT images is proposed, which is simple in structure, possesses good accuracy and faster extraction speed, is easy to integrate into other algorithms to provide a better environment for the subsequent feature extraction of intracranial lesions at a minimal cost of efficiency. Second, the feasibility of combining traditional image processing methods and CNNs to deal with complex segmentation problems is explored. Third, the performance of AMBBEM and 3 FCN models for brain extraction was evaluated, as well as the effect of multiple lesions on various algorithms, which can be used as a reference for related research.

The structure of this paper is as follows. Section 2 presents the data acquisition and processing, as well as the specific ideas of the algorithm. Section 3 presents the arrangement of the experiments and their results. In Section 4, the paper analyzes and discusses the various methods with the experimental results. Finally, Section 5 summarizes the research of this paper and provides an outlook for future work.

## 2. Materials and methods

### 2.1. Data selection and processing

The datasets were derived from the RSNA Intracranial Hemorrhage Original Size PNGs (RIHOSP)dataset, publicly available on the Kaggle website, and the CQ500 dataset on the Academic Torrents website. No patient privacy was involved.

With the assistance of radiologists, we extracted three sets of images from two datasets. First, 1400 slices were selected from the RIHOSP dataset as the first set of images, including 250 at the layer of the basis cranii, 500 at the layer of the sella turcica, and a total of 650 at the remaining layer, some of which contained lesions such as intracranial hematomas and soft tissue edema. Another 140 head CT images were randomly selected from the RIHOSP dataset as the second set of images, with the same proportion of each layer as the first set of images. In the third group, 22 sets of head CT images containing different lesions were screened from the CQ500 dataset, with nine slices in each group. Among the first 18 groups containing lesions, there were 10 cases of intracranial hematoma, 2 cases of each subtype (parenchymal hematoma, ventricular hematoma, subarachnoid hematoma, subdural hematoma, and epidural hematoma); 3 cases of cerebral infarction; 2 cases of soft tissue edema; 2 cases of physiologic calcification; and 1 case of the intracranial cyst. The latter four groups had no lesions, 2 in adults and 2 in minors. The CT slice size was $512 \times 512 \times 3$ without any pre-processing.

The experimental operating system is Windows 11, the processor is AMD Ryzen 7 5700X 8-Core Processor, the graphics card is NVIDIA GeForce 3060Ti, which has 8GB memory for processing data, and the experimental platform is chosen as MATLAB2022b with CUDA version 12.0 (The complete code is given in S1 File).

### 2.2. Algorithm design

The algorithm extracts the brain by multiplying a high-precision mask [19, 20] with the original image. It can be divided into three parts: 1) initial segmentation by threshold segmentation, median filtering, and image filling to generate the mask; 2) the improved ResNet50 model [21] is used to classify the image and then combined with the closed operation to achieve the closure of the skull gap and ensure the complete filling of the mask; 3) according to the classification results, the mask is further trimmed by combining the connected component labeling

method [22], region growing method [23] and image properties analysis to improve the accuracy, and then finally complete the extraction by multiplying the final mask with the original image.

**2.2.1. Preliminary segmentation of brain tissue.** In CT, images of human tissues are formed based on the absorption properties of radiation energy by human tissue [24, 25]. As shown in Fig 1, Fig 1(a) shows the original image, and the skull, pillow, scalp, and accessory tissues are the parts to be removed. Fig 1(b) and 1(c) show the gray value grid surface plot and the gray value (1–254) percentage bar plot of the original image, respectively.

The first peak, d1 in Fig 1(c), corresponds to the gray value distribution of normal brain parenchyma, and the second peak, d2, corresponds to the gray value distribution of cerebral hematoma. Combining Fig 1(b) and 1(c), it can be seen that the gray value of each tissue has a Gaussian distribution, with the skull having the largest gray value at around 255, the intracranial hematoma having the second largest gray value, and the gray value of the brain parenchyma and the extracranial soft tissue are close, both of which are much lower than those of the skull.

In summary, the CT image is first converted into a gray image, and then the skull and brain parenchyma are segmented using threshold segmentation. Considering the influence of CT window width and window level, the threshold range is enlarged to a certain extent. The

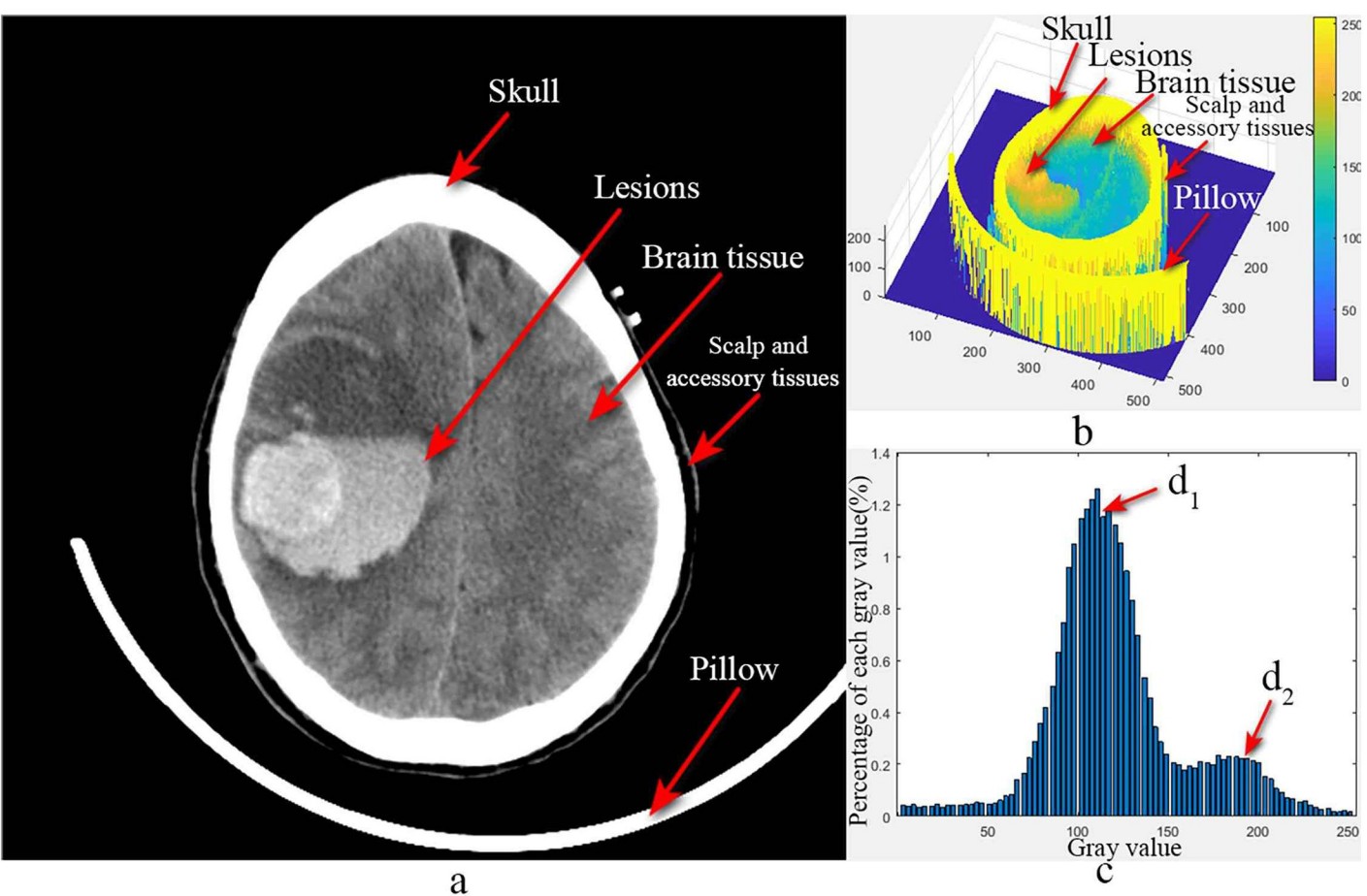

**Fig 1. Analysis of head CT images.** a) Original image b) Mesh surface of the gray value c) Percent bar plot of gray value from 1 to 254.

specific formula is as follows:

$$e_1(i,j) = \begin{cases} 1 & E(i,j) \geq Max(E) - 15 \\ 0 & E(i,j) < Max(E) - 15 \end{cases} \quad (1)$$

$$e_2(i,j) = \begin{cases} e_2(i,j) = E(i,j) & 1 \leq E(i,j) \leq Max(E) - 20 \\ e_2(i,j) = 0 & E(i,j) > Max(E) - 20 \text{ or } E(i,j) < 1 \end{cases} \quad (2)$$

Where $E$ represents the gray image;

$max (E)$ represents the maximum gray value in the gray image;

$e_1$ represents the skull image;

$e_2$ represents the image of skull removal.

The extracted skull is filled as a template to obtain mask 1, and then the image of the removal of the skull is multiplied by mask 1 to remove the skull and extracranial soft tissue. The process is shown in Fig 2.

**2.2.2. Fill detection and skull closure.** As seen above, mask 1 is obtained by filling the segmented skull. As shown in Fig 3(a) and 3(b), the skull is not closed in the head CT images, and there are obvious gaps in the skull after the threshold segmentation, which cannot guarantee the complete filling of the subsequent mask 1. In this paper, the closure of the skull gap is achieved by the closed operation. Considering that the size, location, and shape of the skull gap are changing, cycle 1 is designed in this paper, which is shown in Fig 4. Among them,

$$q = (S_i - S_{e1})/S_i \quad (0 \leq i \leq 5) \quad (3)$$

$i$ represents the number of cycles;

$S_i$ represents the mask area after the $i$ th cycle;

$S_{e1}$ denotes the area of the skull;

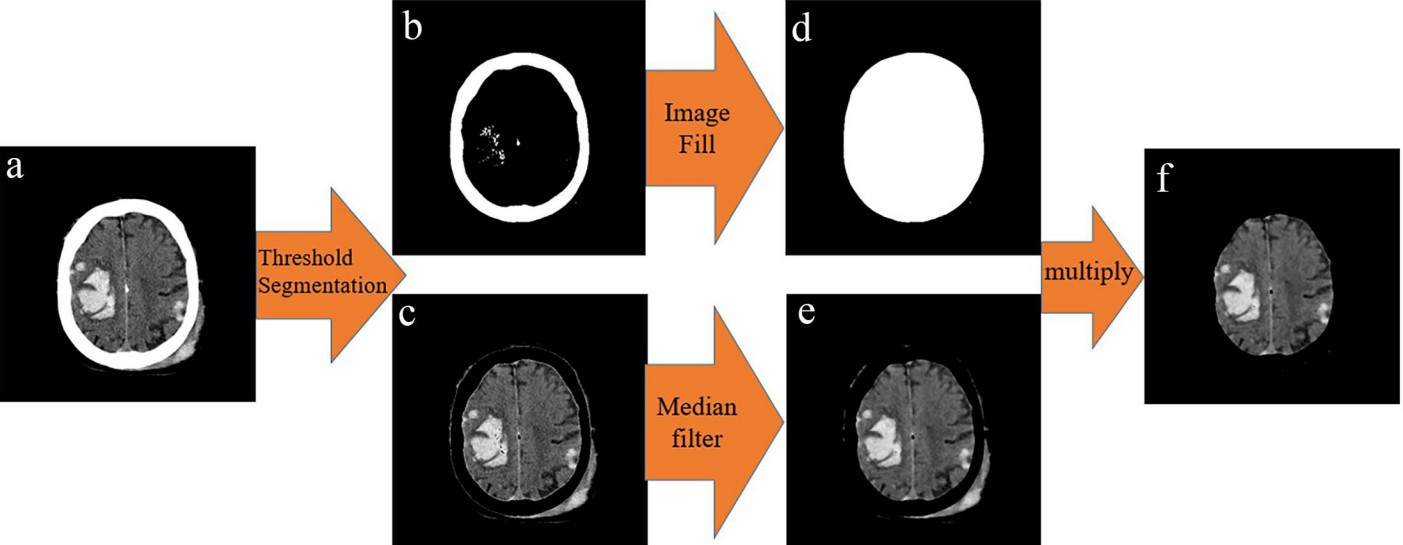

**Fig 2. Diagram of results generated during the initial segmentation process.** a) Original image. b) Skull. c) Image with the skull removed. d) Mask 1. e) Image after noise reduction. f) Initial image with the skull and extra-cranial soft tissue removed.

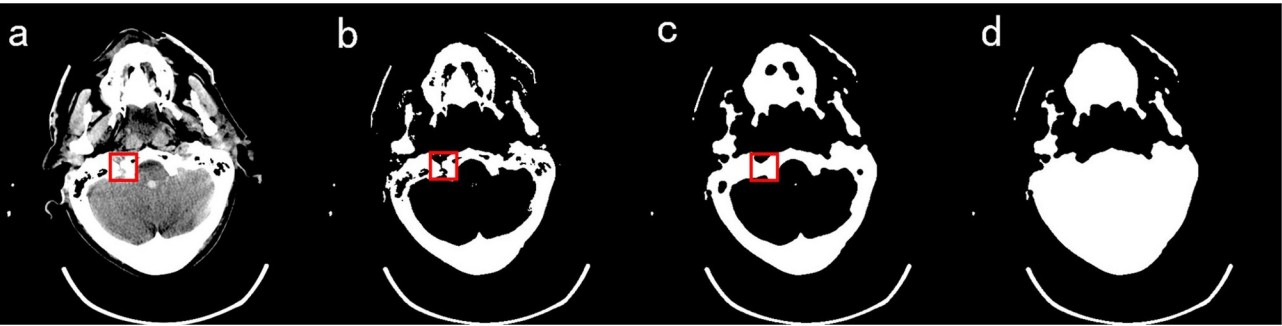

**Fig 3. Diagram of the process of skull closure.** A) Original image b) Skull image c) Images after skull closure d) Image of mask 1.

$S_i$-$S_{e1}$ denotes the filled area of the $i$ th cycle;

$q$ denotes the percentage of the filled area to the mask area after the $i$ th cycle.

The extracted skull is first filled once, and whether the mask is filled completely is judged by whether $q$ is greater than the threshold value (*TV*), where *TV* is obtained by regression fitting from the soft tissue area and the skull area. Observation and regression experiments were performed on 142 groups of head CT images. It was found that the proportion of brain tissue area in the soft tissue area at the basis cranii layer was small, and the fitting effect was poor. The average value of brain area as a percentage of the complete mask was 0.2485. The proportion of brain tissue area in the other layers was larger, and the fitting effect was good, with a

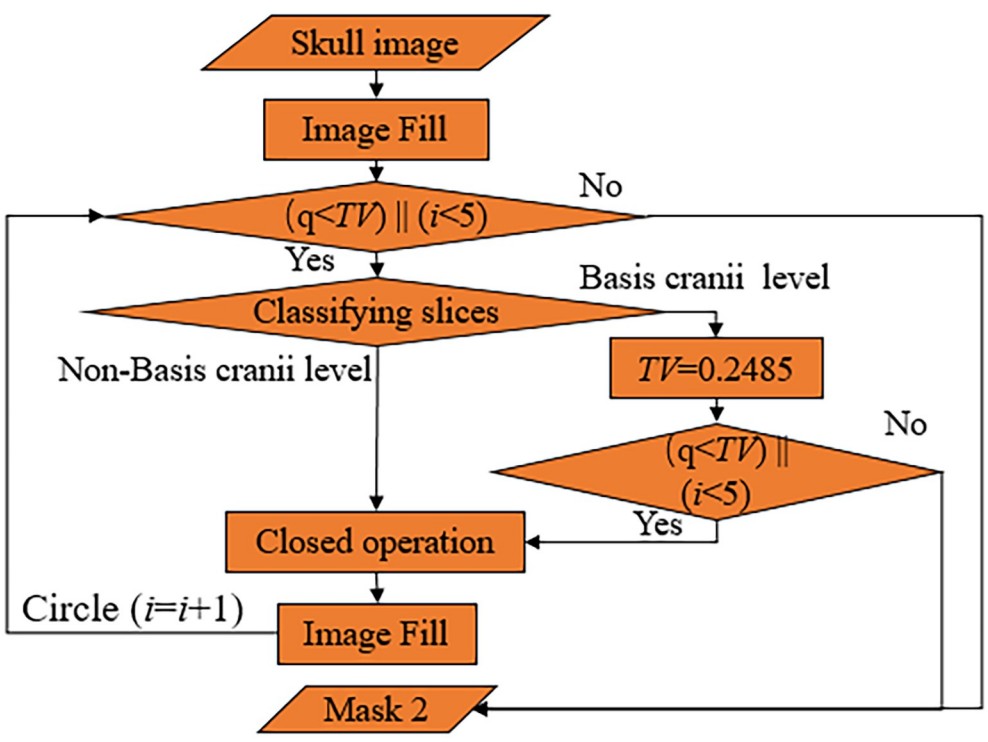

**Fig 4. Cycle 1.**

similarity coefficient of 0.9403 and a p-value of 0 for the statistic, proving that the regression model was established. So when $q$ is less than $TV$, the convolutional neural network(CNN) discriminates the image. If it belongs to the basis cranii layer, $TV$ is reassigned to 0.2485, after which the relationship between $q$-value and $TV$ is judged again, and if it belongs to other layers, closed operation and refilling are performed directly. The structural element of the closed operation increases one by one during the cycle. If $q$ is greater than $TV$, the cycle is jumped out, and the following steps are continued. It is also observed that a small portion of images with a small brain area exists, and even if a complete filling is obtained, $q$ is still less than this $TV$. To avoid falling into a dead cycle, the maximum number of cycles is limited at the same time. The closed operation is tested on 142 sets(A total of 3132 sheets) of head CT images. When the number of cycles reaches 3, all the images containing gaps are closed, but the increase in the number of closed operations will bring errors to the subsequent segmentation, and we set the maximum number of cycles to 5 in careful consideration. As shown in Fig 3(c), the gap of the skull is closed. After filling, mask 1 is obtained, as shown in Fig 3(d).

We used five CNN models to classify the original images, classifying the images into a total of four classes, with the basis cranii layer as a separate class and the other layers divided into three classes based on the number of brain distribution regions. The first three networks are VGG19 [26], EfficientNet [27], and RestNet50. The fourth one is an improved version of the RestNet50 network, which improves RestNet50 in two ways: firstly, adding a Convolutional Block Attention Module (CBAM) [28] to the Stem Block section of ResNet50 and then adding Squeeze Excitation (SE) [29] to each of the four Bottleneck sections. Squeeze Excitation). The first four networks have an input layer size of $224 \times 224 \times 3$, which is converted to $224 \times 224 \times 3$ using bilinear interpolation [30] before the image is input. In the end, the softmax function is used to generate four types of outputs. The fifth one is the SqueezeNet network [31], where the size of the input layer of the network is $227 \times 227 \times 3$. The image is transformed using bilinear interpolation before the image input, and the number of channels in the final convolutional layer is adjusted to four.

**2.2.3. Re-segmentation of the mask.** In the whole set of head CT sections, some sections are relatively more complex in structure, and it is difficult to ensure high-quality brain extraction by preliminary segmentation only, as shown in Fig 5(a). Through preliminary segmentation, non-brain tissues are still not removed and need further detection and segmentation. We first perform median filtering on the initially segmented image to remove small areas of non-brain tissue, producing mask 2, as in Fig 5(b). Then, the connected component labeling method measures the number of connected regions of the mask. If the number of connected regions equals 1, mask 2 is used as the final mask. If it is greater than 1, different methods are used to re-segment mask 2 according to CNN's different classification results.

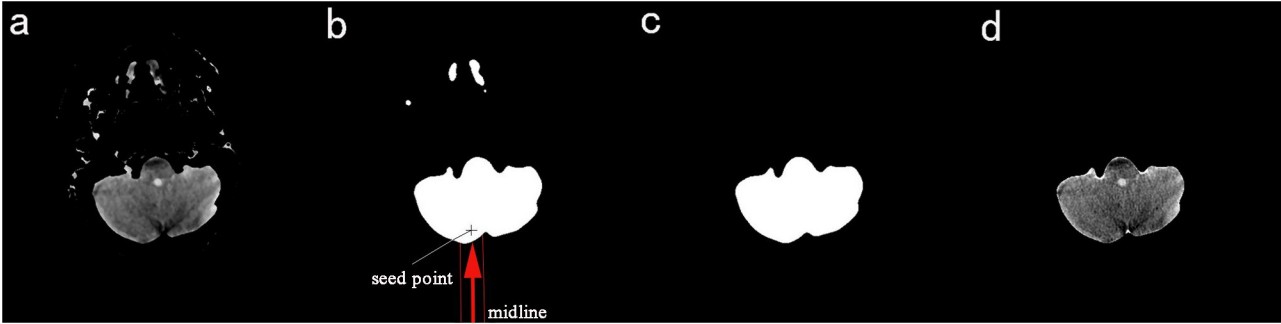

**Fig 5. Process diagram for mask re-segmentation.** a) Initial segmented image b) Image of Mask 2 c) Final mask image d) Final brain extraction image.

In a set of head CT images, due to the presence of the skull, the brain tissue in some of the images is distributed as a single block, and some of them are distributed in multiple regions. We classify the head CT images into brain single-region distribution and brain multi-region distribution according to the difference in brain tissue distribution. As shown in Fig 6, for the images that require mask re-segmentation, we first use CNN to discriminate. For the images with the single-region distribution of the brain, we use the region growing [32] algorithm to trim the clipping of the mask 2. After much observation, In mask 2, the part to be eliminated originates from the human tissue above the brain parenchyma and not below the brain parenchyma, and the previous steps have eliminated the skull below the brain parenchyma and the extracranial soft tissues. Moreover, the brain is distributed in the middle of the image. Therefore, the first point that is not '0' can be used as the seed point for the region growth algorithm by searching from bottom to top within a certain range of the image midline. To ensure the robustness of segmentation, we moved the position of this point up another five lines to ensure that the seed point can fall precisely in the target region, see Fig 5(b) for details, and the seed point falls accurately in the region corresponding to the brain parenchyma. After capturing the seed points, the region growing algorithm is used to realize the re-segmentation of mask 2, and then the final mask is obtained, as in Fig 5(c). After that, the original image is multiplied with the final mask, and then the brain parenchyma extraction is completed, as in Fig 5(d).

For images with the multi-regional distribution of the brain, using the above methods results in small areas of missing brain tissue, so we perform segmentation based on image properties. The image is first transformed into a binary image. Then, the mask is re-segmented with the area size (number of pixels in each region) as the specified attribute and the classification result of CNN as the number of extractions.

In summary, the overall flow chart of the AMBBEM algorithm is shown in Fig 7. The soft tissue and skull are separated by threshold segmentation first, and then the image is filled with the skull as the template to get mask 1. The fill detection and closed operations are designed to overcome the problem of skull gaps, further ensure the fill integrity of mask 1, and increase the algorithm's robustness. In the whole algorithm process, median filtering can eliminate the small area of non-brain tissue in the image. The connected component labeling method determines whether further segmentation of the mask is needed. The re-segmentation of the mask was done to improve further the accuracy of the final segmentation, details of which are shown in Fig 6 above. In addition, CNN is used throughout the algorithm to classify images that require closed operations and re-segmentation, providing a basis for selecting the appropriate

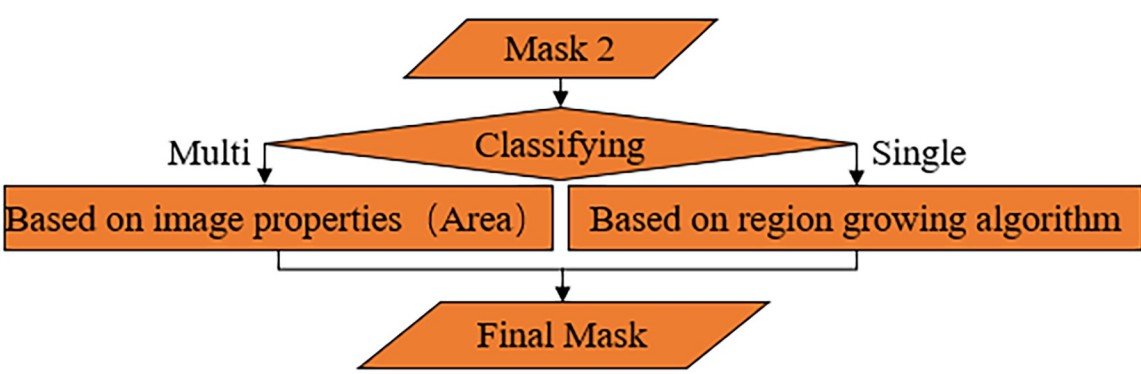

**Fig 6. Mask re-segmentation.**

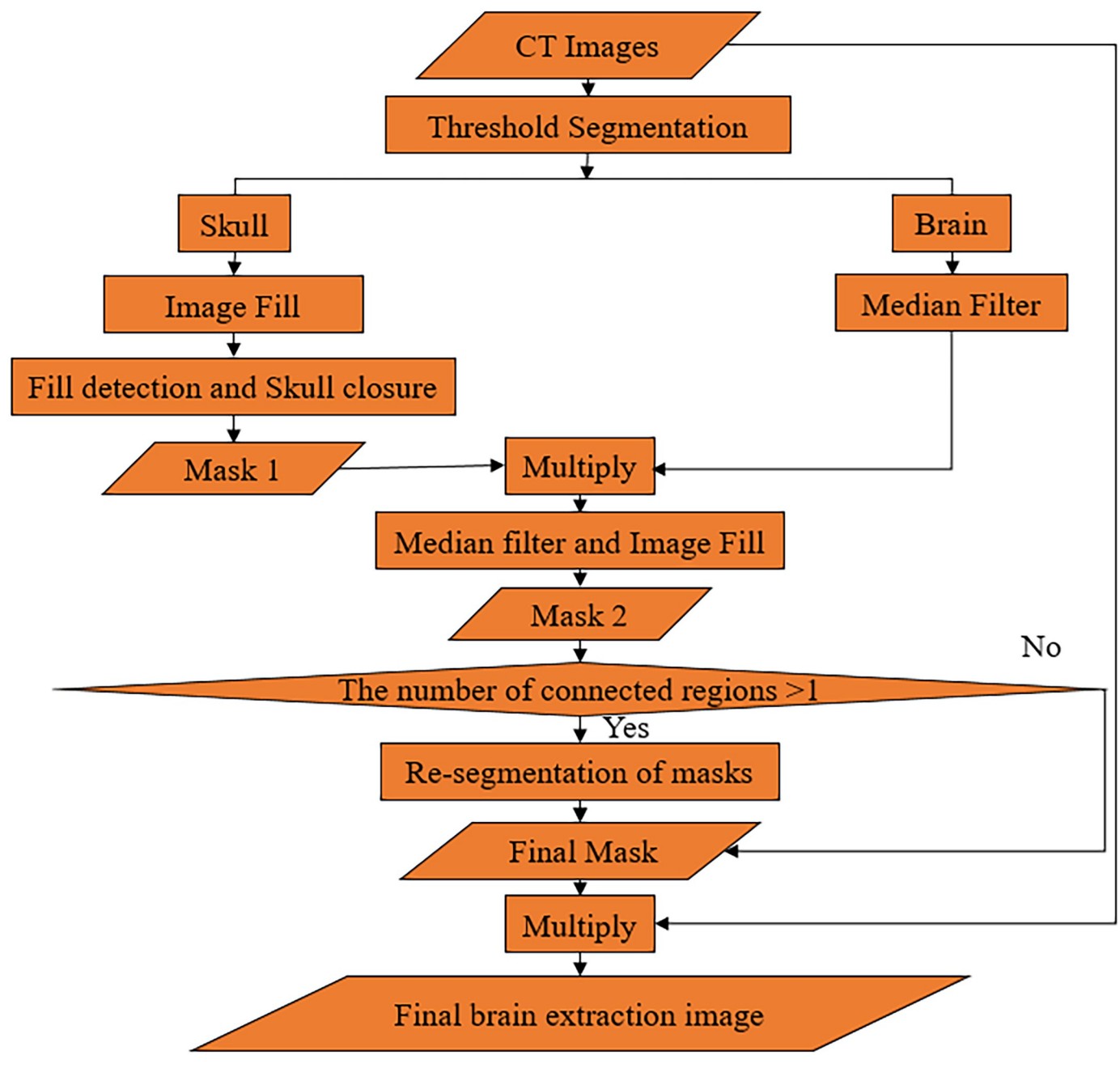

**Fig 7. Flowchart of AMBBEM.**

processing method for different images and largely increasing the robustness and accuracy of the algorithm.

## 3. Experiment

To verify the AMBBEM algorithm's robustness, accuracy, and segmentation speed, this paper compares the algorithm with three FCN models: Deeplabv3+ [33], U-net [34], and SegNet [35]. The first set of images is used as the training set for the six CNN models and the three

FCN models (The training set for CNNs is given in S1 Data, the training set of FCN is given in S2–S4 Data), the second set of images is used as test set 1 for testing the classification effect of the six CNN models (The test set 1 is given in S5 Data), and the third set of images is used as the test set 2 for testing the brain extraction effect of AMBBEM and the three FCN models (The test set 2 is given in S6 Data).

### 3.1. Training settings

During the training of the five CNN models, the initial learning rate is 0.001, and the BatchSize is 16. To determine the best epochs, we first trained with 15 epochs and then increased the number of epochs by 15 each time and made observations. As shown in Fig 8, it is observed that the training loss of each CNN model tends to be stable at epochs of 30, so we finally set the

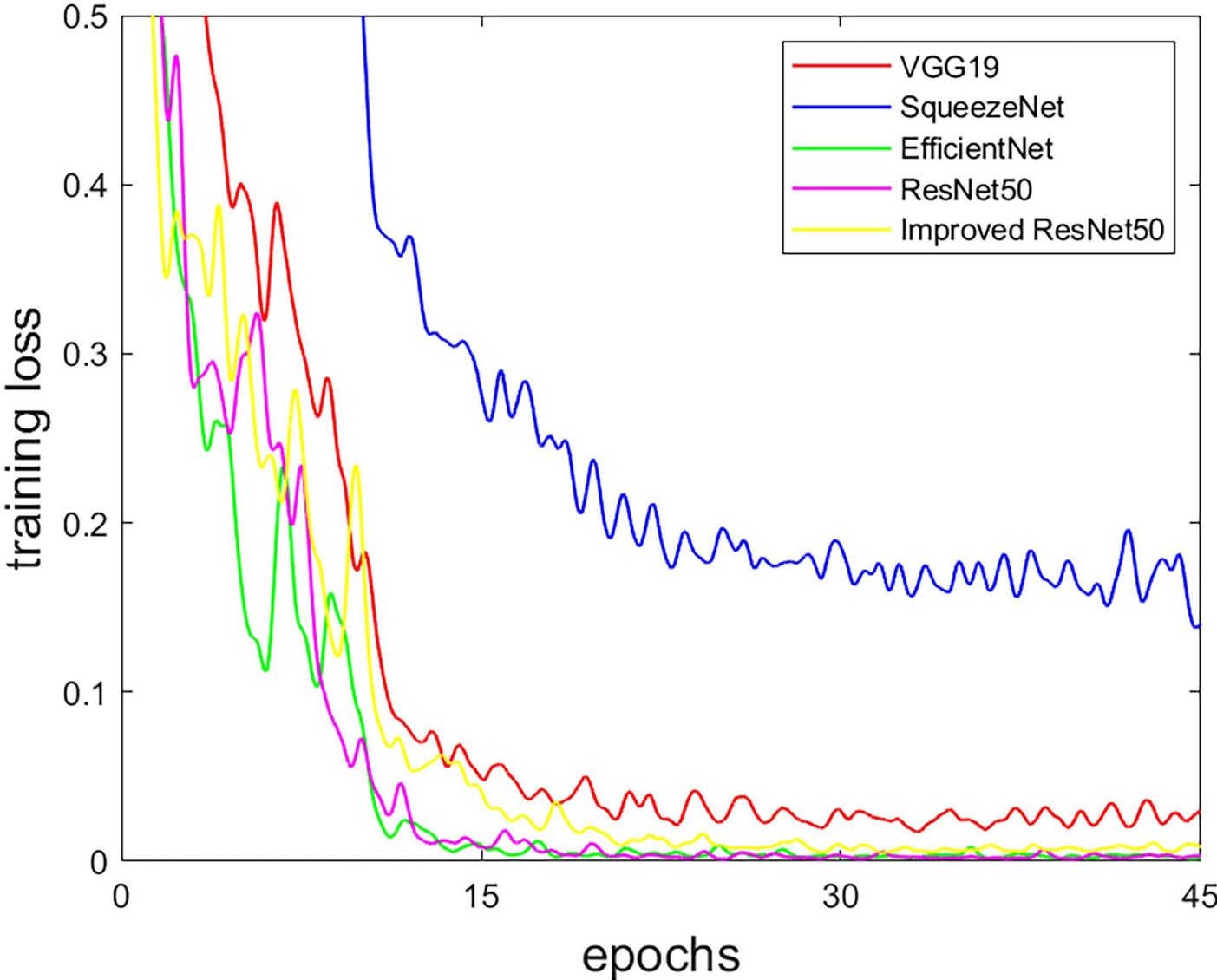

**Fig 8. The training loss curve of each CNN.**

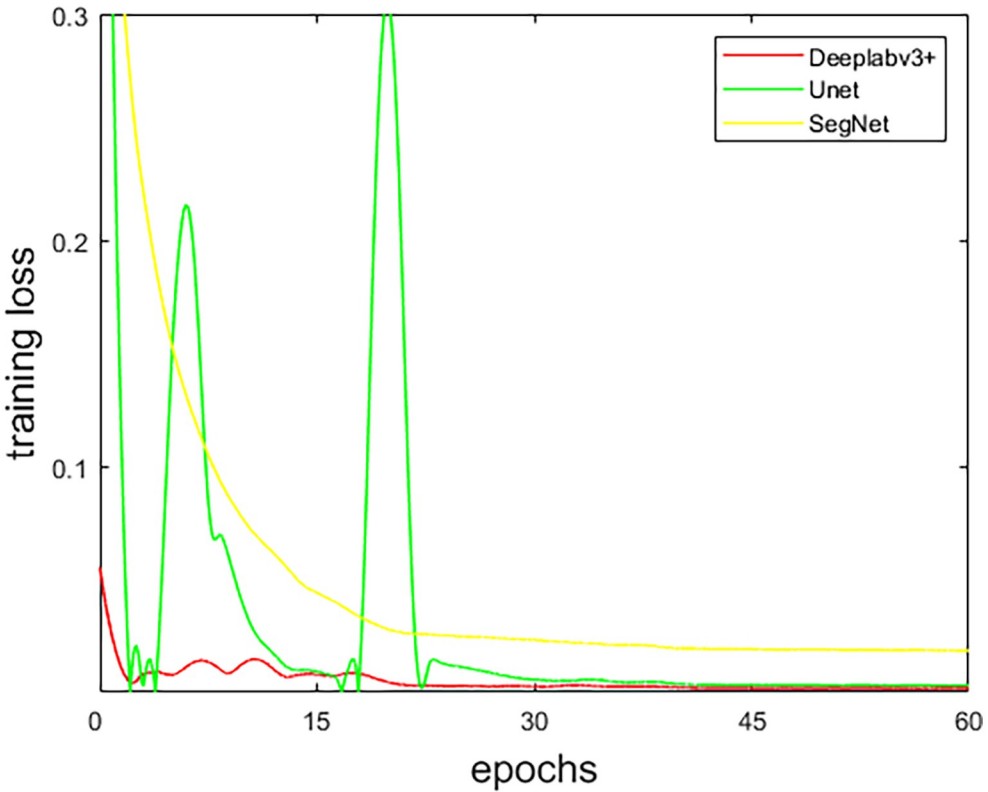

**Fig 9. The training loss curve of each FCN.**

epochs to 45. In order to determine the best optimizer, each CNN model is tested simultaneously with three optimizers, Adam, SGD, and RMSProp, and the one with the best test results is selected. The optimizer for VGG19 is SGD; the optimizer for ResNet50, EfficientNet, and the improved ResNet50 is Adam; and SqueezeNet is the RMSProp optimizer.

Similarly, during the training of the three FCN models, after comparison, the optimizer chooses Adam with an initial learning rate of 0.001, and the BatchSize is 8. Also, in Fig 9, it is observed that the training loss of the three FCN models tends to be stable when the epochs are 45, so we finally set the epochs to 60.

### 3.2. Evaluation indicators

Among the five CNN models tested, this paper chooses two standard metrics, accuracy and average precision (AP), as evaluation criteria. In the final test of AMBBEM and three FCN models, Mean Pixel Accuracy (MPA), Mean Intersection over Union (MIoU), Mean boundary F1-Measure Score (MBF) [36], and Average speed of segmentation (Ass) are selected as evaluation metrics.

As shown in Table 1, where TP (true positive) indicates the number of brain tissue(BT) predicted as BT in all pixels of the cranial CT images, FP (false positive) shows the number of BT predicted as non-brain tissue (NBT) in the pixels of the cranial CT images; TN (true negative) indicates the number of NBT predicted as NBT and FN (false positive) indicates the number of NBT expected as BT. The MPA is used to represent the average of the BT and NBT accuracy

**Table 1. Dichotomous confusion matrix.**

| True Results | Predicted Results | |
|---|---|---|
| | **Positive Sample** | **Negative Sample** |
| **Positive Sample** | TP | FN |
| **Negative Sample** | FP | TN |

and is calculated as follows:

$$MPA = \frac{1}{2}\left(\frac{TP}{TP + FP} + \frac{TN}{TN + FN}\right) \tag{4}$$

In the semantic extraction, IoU (Intersection over Union) denotes the interaction rate between the prediction frame and the real frame, while the MIoU denotes the average value of the IoU between BT and NBT, so the MIoU is calculated as:

$$MIoU = \frac{1}{2}\left(\frac{TP}{TP + FP + FN} + \frac{TN}{TN + FN + FP}\right) \tag{5}$$

Also, considering the importance of boundary accuracy in medical image extraction, this paper used the F1-measures as an evaluation index, using 0.75% of the image's diagonal as the tolerance distance. MBE denotes the average of the F1-Measure of the BT and NBT. The Ass then indicates the number of sheets extracted per second and is tested as follows:

$$Ass = N/T \tag{6}$$

Where N denotes the total number of images, T denotes the whole time to complete the segmentation, test set 2 is used as the test object and tested five times, and the results are averaged.

## 3.3. Results

Table 2 shows the classification results of the five CNN models on Test Set 1. Among them, the improved ResNet50 has the highest accuracy and AP of 99.31% and 99%, which are 2.07% and 2.44% higher than the original ResNet50, 3.45%, and 3.88% higher than VGG19, and 6.90% and 8.22% higher than EfficientNet, respectively, while the SqueezeNet performs relatively poorly, much lower than the other four network models. Taken together, we chose the improved ResNet50 for the task of head CT image recognition.

Table 3 shows the performance of AMBBEM and the three FCN models in Test Set 2. Among the four evaluation metrics, the MPA value of the AMBBEM algorithm is comparable to the SegNet model, which is 0.10% higher than the U-net model but 0.03% lower than Deeplabv3+. The MIoU value is highest with the AMBBEM algorithm, which is 0.10%, 0.14%, and 0.17% higher than the Deeplabv3+, U-net, and SegNet models, respectively. The MBF value of the AMBBEM algorithm is 0.10% lower than that of the Deeplabv3+ model but 0.62% and

**Table 2. Performance of each CNN model in test set 1.**

| | VGG19 | SqueezeNet | EfficientNet | ResNet50 | Improved ResNet50 |
|---|---|---|---|---|---|
| **Accuracy** | 0.9586 | 0.8483 | 0.9241 | 0.9724 | 0.9931 |
| **AP** | 0.9512 | 0.8229 | 0.9078 | 0.9656 | 0.9900 |

**Table 3. Performance of AMBBEM and the three FCN models in test set 2.**

|  | MPA | MIoU | MBF | Ass(Sheets/second) |
|---|---|---|---|---|
| AMBBEM | 0.9963 | 0.9924 | 0.9941 | 6.160 |
| Deeplabv3+ | 0.9966 | 0.9914 | 0.9951 | 0.585 |
| U-net | 0.9953 | 0.9910 | 0.9879 | 0.581 |
| SegNet | 0.9963 | 0.9907 | 0.9934 | 0.587 |

0.07% higher than that of the U-net and SegNet models, respectively. The Ass of the AMBBEM algorithm is 6.16 sheets/second, which can complete the brain extraction of one head CT image in about 0.16 seconds. It is 10.53 times faster than Deeplabv3+, 10.6 times faster than U-net, and 10.49 times faster than the SegNet model, respectively.

In a group of head CT images, the images at the basis cranii and sella turcica layers are more complex, and an unclosed skull is common. In addition, the brain area is smaller in the basis cranii layers images, while the other soft tissue areas are larger. In the images of the sella turcica layers, the brain is distributed in multiple regions. To further test the characteristics of each algorithm, images at the basis cranii layers, sella turcica layers, and other layers in test set 2 were examined separately. As shown in Tables 4–6, the test results of the AMBBEM and the three FCN models at the basis cranii layers, the sella turcica layers, and other layers, respectively, are shown. The segmentation effect of the AMBBEM algorithm in the basis cranii layer is significantly better than that of the three FCN models, but in the segmentation test at the sella turcica level, the segmentation effect of the AMBBEM algorithm is significantly lower

**Table 4. Test results of each algorithm at the basis cranii layers.**

|  | MPA | MIoU | MBF |
|---|---|---|---|
| AMBBEM | 0.9925 | 0.9842 | 0.9909 |
| Deeplabv3+ | 0.9915 | 0.9753 | 0.9776 |
| U-net | 0.9874 | 0.9754 | 0.9696 |
| SegNet | 0.9909 | 0.9713 | 0.9741 |

**Table 5. Test results of each algorithm at the sella turcica layers.**

|  | MPA | MIoU | MBF |
|---|---|---|---|
| AMBBEM | 0.9848 | 0.9759 | 0.9808 |
| Deeplabv3+ | 0.9929 | 0.9827 | 0.9930 |
| U-net | 0.9902 | 0.9824 | 0.9887 |
| SegNet | 0.9924 | 0.9819 | 0.9929 |

**Table 6. Test results of each algorithm at the other layers.**

|  | MPA | MIoU | MBF |
|---|---|---|---|
| AMBBEM | 0.9971 | 0.9938 | 0.9964 |
| Deeplabv3+ | 0.9970 | 0.9924 | 0.9973 |
| U-net | 0.9959 | 0.9920 | 0.9901 |
| SegNet | 0.9967 | 0.9918 | 0.9958 |

than that of the three FCN models. In other layers of testing, AMBBEM extracts segmentation results similar to Deeplabv3+ and slightly better than U-net and SegNet models.

Figs 10 and 11 shows the extraction effects of AMBBEM and three FCN models at the basis cranii and sella turcica layers (All segmentation effect images for AMBBEM and FCN in Test Set 2 are detailed in S1 Fig). As shown in Fig 10, in the randomly selected eight images, the three models of Deeplabv3 +, U-net, and SegNet all showed a small area of extracranial soft tissue that was not removed, and no similar situation was found in the test of the AMBBEM algorithm. However, in the extraction effect map of the sella turcica layer, the AMBBEM algorithm showed a small area of brain tissue loss, while no loss was found in the extraction effect map of the three FCN models.

Fig 12 shows the effect of various methods on extracting images containing different lesions. The U-net model has a small area deletion in the area of intracranial hematoma and cerebral infarction. However, AMBBEM, Deeplabv3+, and SegNet were unaffected by various lesions and showed good robustness.

## 4. Discussion

We propose an adaptive mask-based brain extraction method for head CT images. Threshold segmentation, region growing, and improved ResNet50 are skillfully combined by designing a special detection mechanism, a loop structure, and an automatic seed point selection method. The method can accomplish brain extraction of approximately 6 CT images in 1 second using NVIDIA GeForce 3060Ti.

For image segmentation speed, traditional image processing methods such as tracking algorithms, WMFCM, FCM, threshold segmentation, DRLSE, etc., are close to AMBBEM. However, when facing the problems of extracranial soft tissue edema, unclosed skull, and multi-region distribution of the brain, the above traditional image processing methods may have

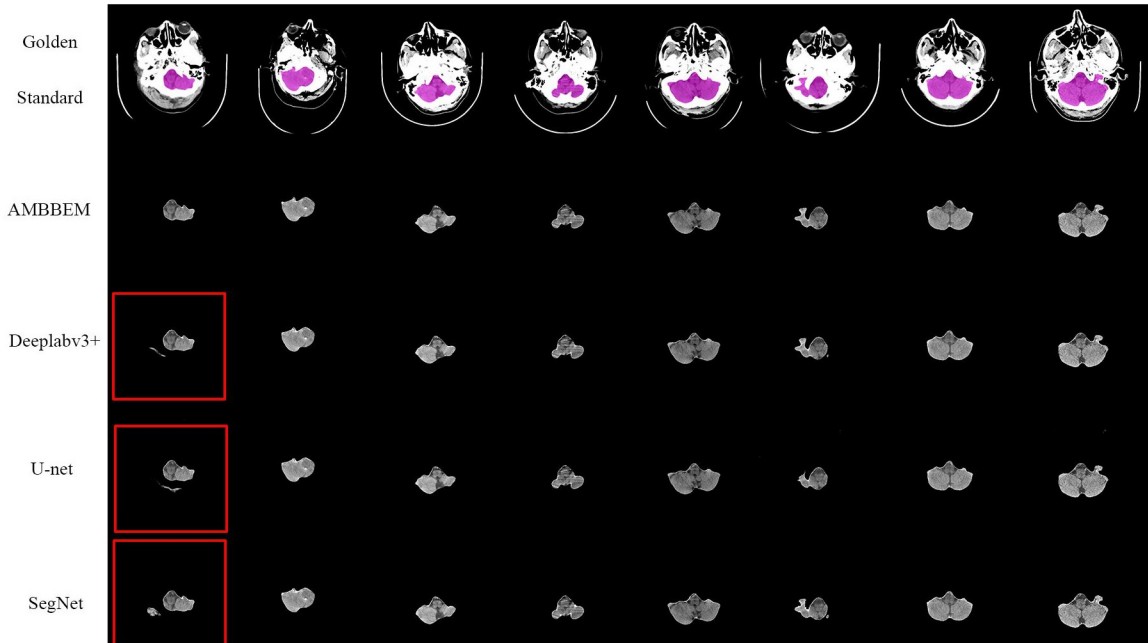

**Fig 10. Extraction performance of AMBBEM and three FCN models at the basis cranii layer.**

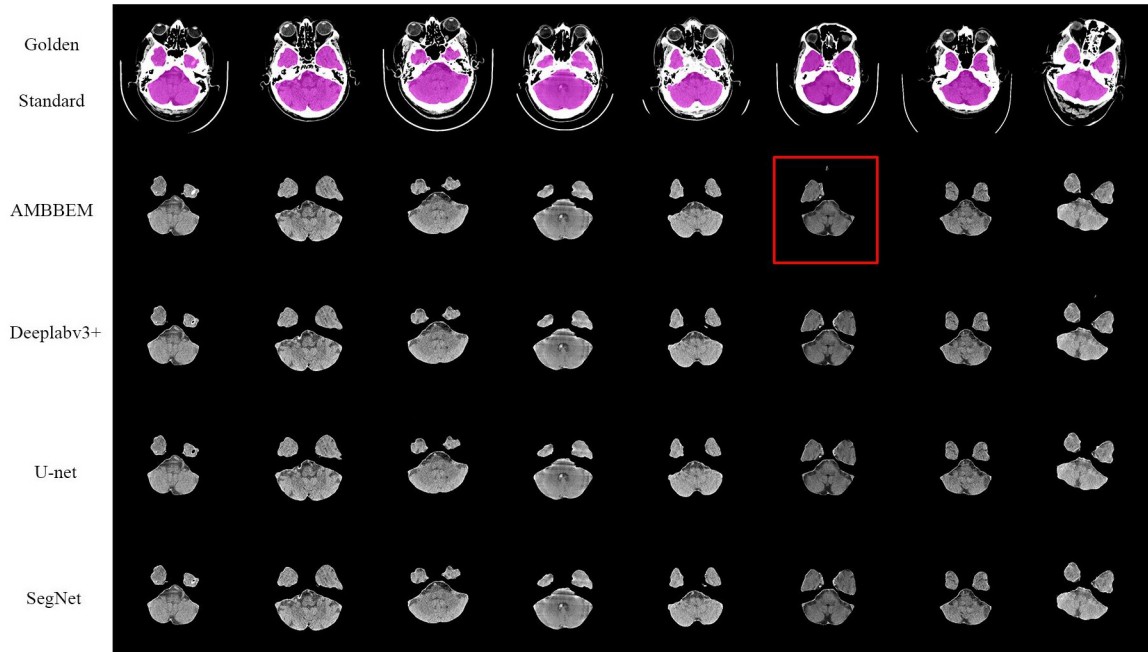

**Fig 11. Extraction performance of AMBBEM and three FCN models at the sella turcica layer.**

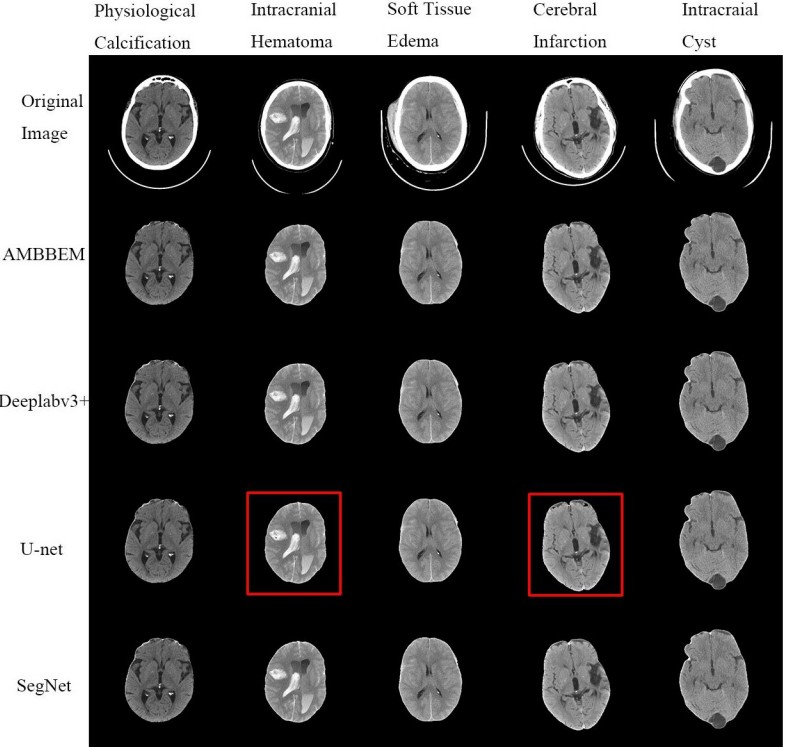

**Fig 12. Extraction effect of each algorithm on images containing different lesions.**

certain limitations, such as non-brain tissue not being excluded and brain tissue loss. However, AMBBEM can effectively overcome the above problems with better robustness and accuracy and can be applied to extract the brain from the whole set of head CT images.

The secondary development of medical image post-processing software has a decent segmentation accuracy but relies on a certain software platform. It is difficult to integrate the function of brain extraction into other systems separately. AMBBEM, on the other hand, with its simple structure, is easier to combine with related algorithms.

The FCN model has both good robustness and accuracy in the task of brain extraction from CT images. Compared with the FCN model, the segmentation accuracy of AMBBEM is close to that of FCN, but the segmentation speed is much faster than that of the FCN model, which is several times its segmentation speed. At the same time, the FCN model suffers from the need for big data and black box problems, while AMBBEM is easy to implement and has better interpretability due to its relatively simple structure and principle.

For the classification task of head CT images, the improved ResNet50 significantly outperforms the original ResNet50 as well as other networks, which proves the effectiveness of the improved method. This is due to the fact that the SE module and CBAM module multiply the learned weights with the input feature maps and adaptively adjust the weights of the feature maps, which improves the quality and diversity of the features and motivates the network to capture the key information in the images better.

Frankly speaking, our algorithm also has some shortcomings. There is a certain probability of missing brain tissue in the segmentation task of the sella turcica layer image. This is caused by the simultaneous presence of multiple gaps in the skull. When multiple gaps exist, AMBBEM performs a closed operation due to $q$ being less than $TV$, but when one of the gaps is closed, subsequent image filling increases $q$ and thus exceeds $TV$, but at this time, the other gaps are not closed, which results in the loss of a small area of brain tissue.

## 5. Conclusion

The AMBBEM proposed in this paper combines traditional image processing methods with CNN, which can accomplish 6.16 images per second while ensuring similar accuracy to the FCN model, reaching tens of times the latter. The feasibility and superiority of combining traditional image processing methods with CNN to solve complex segmentation tasks quickly is demonstrated. The corresponding contribution is to provide a fast and accurate brain extraction method that can replace manual segmentation to realize brain extraction from head CT images, which in turn provides the basis for automated measurement of brain volume and automated detection of intracranial lesions. In our future work, we will try to combine this method with FCN to improve the accuracy and robustness of the algorithm further by giving the task of segmenting the sella turcica layer to FCN. As well as expanding the scope of testing in order to find problems and then further adjust the algorithm.

## Supporting information

**S1 File. This is the program code.**
(ZIP)

**S1 Data. This is the training set for the CNN.**
(ZIP)

**S2 Data. This is the label of the FCN training set.**
(ZIP)

**S3 Data. This is the image of the FCN training set.**
(ZIP)

**S4 Data. This is the image of the FCN training set.**
(ZIP)

**S5 Data. Test set 1.**
(ZIP)

**S6 Data. Test set 2.**
(ZIP)

**S1 Fig. This is the segmentation effect of AMBBEM and FCN in test set 2.**
(ZIP)

## Author Contributions

**Conceptualization:** Dingyuan Hu.

**Data curation:** Dingyuan Hu.

**Formal analysis:** Chunyu Han.

**Funding acquisition:** Hongbin Liang.

**Investigation:** Yuhang Jiang.

**Methodology:** Dingyuan Hu.

**Project administration:** Hongbin Liang.

**Resources:** Dingyuan Hu.

**Software:** Dingyuan Hu.

**Supervision:** Hongbin Liang, Qingyan Zhang.

**Validation:** Dingyuan Hu.

**Visualization:** Dingyuan Hu.

**Writing – original draft:** Shiya Qu.

**Writing – review & editing:** Shiya Qu.

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
