## [Decision Letter · Decision Letter 0]

4 Jun 2023

PONE-D-23-11044Adaptive Mask-Based Brain Extraction Method for Head CT ImagesPLOS ONE

Dear Dr. Liang,

Thank you for submitting your manuscript to PLOS ONE. After careful consideration, we feel that it has merit but does not fully meet PLOS ONE’s publication criteria as it currently stands. Therefore, we invite you to submit a revised version of the manuscript that addresses the points raised during the review process.

Thank you to the reviewers for their valuable comments and evaluations of the manuscript titled "Adaptive Mask-Based Brain Extraction Method for Head CT Images." After carefully considering the reviews and conducting my own evaluation, I have decided to request major revisions before accepting the manuscript for publication. The reviewers have identified several areas that require significant improvements to enhance the clarity, methodology, and comparison with other methods.

We look forward to receiving your revised manuscript.

Kind regards,

Khan Bahadar Khan, Ph.D

Academic Editor

PLOS ONE

“This research was supported by the Department of Science and Technology of Liaoning Province, Natural Foundation Project (No: 2015020128).”

“This research was supported by the Department of Science and Technology of Liaoning Province, Natural Foundation Project (No: 2015020128).”

“This research was supported by the Department of Science and Technology of Liaoning Province, Natural Foundation Project (No: 2015020128).”

5. Please upload a copy of Supporting Information S1 File which you refer to in your text on page 10.

Reviewers' comments:

Reviewer's Responses to Questions

**Comments to the Author**

1. Is the manuscript technically sound, and do the data support the conclusions?

Reviewer #1: Yes

Reviewer #2: Yes

2. Has the statistical analysis been performed appropriately and rigorously? 

Reviewer #1: Yes

Reviewer #2: Yes

3. Have the authors made all data underlying the findings in their manuscript fully available?

Reviewer #1: Yes

Reviewer #2: Yes

4. Is the manuscript presented in an intelligible fashion and written in standard English?

Reviewer #1: No

Reviewer #2: No

5. Review Comments to the Author

Reviewer #1: Dear editor

The paper “ Adaptive Mask-Based Brain Extraction Method for Head CT Images” contains the extract brain parenchyma from head CT images and they used the different computerized techniques. This paper could be published in this journal after some major comments for improving the paper.

1.The introduction section should be re-write based on the introduction of the field as well motivation and problem statement as well as contribution of the work. The authors can get idea from the following papers.

“ Enhancement of Medical Images through an Iterative McCann Retinex Algorithm: A Case of Detecting Brain Tumor and Retinal Vessel Segmentation YE Almalki, NA Jandan, TA Soomro, A Ali, P Kumar… - Applied Sciences, 2022. “

“Image segmentation for MR brain tumor detection using machine learning: A Review”, TA Soomro, L Zheng, AJ Afifi, A Ali, S Soomro, M Yin… - IEEE Reviews in Biomedical Engineering, 2022.

“Neural network based denoised methods for retinal fundus images and MRI brain images”, TA Soomro, J Gao - 2016 International Joint Conference on Neural …, 2016.

2.A related work section should be introduced.

3.The proposed method is the combination of multiple existing methods , author should explain the purpose of use these methods in this research along with their contribution.

4.The results section should be improved and compared with other recently proposed techniques and why only CT images , why do they not use MRI images also. ?

Reviewer #2: An algorithm based on adaptive masking is proposed to extract brain parenchyma in CT images, combined with traditional image processing and the AlexNet network. It outperforms U-net and Deeplabv3+ models, achieving better MPA, MIoU, and MBF and faster speed. It is significant to practical applications. However, there are a few more comments about the presentation and writing.

Major comments:

1. The AlexNet is a classic and sound network, but not perfect today. Have you ever tried some relatively SOTA classification networks?

2. The Alexnet is improved by batch normalization. How about the Unet using batch normalization?

3. It is unfair that the AlexNet network used 5000 images for training while U-net and Deeplabv3 used 1400 images.

4. In experiments, the training settings, such as optimizer, learning rate, BatchSize, epochs et al., are identical for Unet and Deeplabv3+. Do they both converge and get the best results?

5. Although the Unet network is widely used in medical image segmentation, it is not designed for segmenting CT brain images. Have you ever compared the proposed method with other methods?

Minor comments:

1. Figure 6 is upside down, and "The algorithm designs a second loop as shown in Figure 7" actually means in Figure 6.

2. All Figure numbers are mismatched in the text from Figure 6.

3. Lack of explanation for some settings, such as why cycle 1 is 5 times instead of 3 or 10 times, and Q is 0.22, and so on. Moreover, the results of Alexnet with batch normalization or without are not compared.

4. Some figures and formulas are difficult to read. Highlight the best results of each evaluation metric in tables.

5. Some sentences are hard to follow, so language and grammar editing is highly recommended.

6. PLOS authors have the option to publish the peer review history of their article (what does this mean?). If published, this will include your full peer review and any attached files.

Reviewer #1: No

Reviewer #2: No

---

## [Author Response · Author response to Decision Letter 0]

1 Jul 2023

Dear Editors,

Many thanks to the editor and reviewers for their valuable comments. We have adjusted our work accordingly based on your comments and those of the two reviewers, and the adjustments are shown in ' Revised Manuscript with Track Changes,' 'Manuscript,' and 'Cover Letter, ' respectively, according to your request. Below we detail the changes we have made to each of the recommendations.

First, for the five editorial requirements, we made the following adjustments:

Our response： We apologize for the formatting issues with the manuscript. We have revised the paper according to the PLOS ONE style template. We refer to the latest published articles in PLOS ONE for some unsure details of the modification. If formatting issues still need to be revised, please give us your specific suggestions, and we will make modifications as soon as possible according to your requirements.

2. Please note that PLOS ONE has specific guidelines on code sharing for submissions in which author-generated code underpins the findings in the manuscript. In these cases, all author-generated code must be made available without restrictions upon publication of the work. 

Our response： We understand and support the journal’s requirements and are willing to actively cooperate with the related work. The source code, dataset, and mathematical model are put into the supporting information. If specific operations are required in the future, we will actively cooperate.

3. Please state what role the funders took in the study.

Our response: Based on the actual situation, we provide a more detailed description of the funders’ situation in the cover letter.

Our response: We removed the funding information from the manuscript and added a statement to the cover letter upon request.

5. Please upload a copy of Supporting Information S1 File which you refer to in your text on page 10.

Our response: We apologize for failing to upload support information in time! We have reorganized the support information and uploaded them one by one. If there is still a need for additional information, we will be sure to upload it upon request.

In response to reviewer #1's four comments, we made the following changes:

1. The introduction section should be re-write based on the introduction of the field as well motivation and problem statement as well as contribution of the work. The authors can get ideas from the following papers. “ Enhancement of Medical Images through an Iterative McCann Retinex Algorithm: A Case of Detecting Brain Tumor and Retinal Vessel Segmentation” YE Almalki, NA Jandan, TA Soomro, A Ali, P Kumar… - Applied Sciences, 2022.

“Image segmentation for MR brain tumor detection using machine learning: A Review,” TA Soomro, L Zheng, AJ Afifi, A Ali, S Soomro, M Yin… - IEEE Reviews in Biomedical Engineering, 2022.

“Neural network based denoised methods for retinal fundus images and MRI brain images,” TA Soomro, J Gao - 2016 International Joint Conference on Neural …, 2016.

Our response: First of all, we acknowledge and appreciate the reviewer's suggestion and realize the shortcomings of the introduction section and that drawing on this suggestion does make the introduction section more organized. Therefore, we read the recommended articles in detail according to the reviewers’ suggestions and used some of them for reference. Finally, we have rewritten the introduction according to the proposal to provide an introduction to the current state and problems of the field and to describe specifically the motivation for our work and the corresponding contributions.

2. A related work section should be introduced.

Our response According to this recommendation, we provide an overview of the relevant literature and studies in the introduction, outlining in detail the implications, gaps, and problems of the existing work. And in the discussion section of the article, we position our study.

3. The proposed method is the combination of multiple existing methods; author should explain the purpose of use these methods in this research along with their contribution.

Our response: After reading this comment, we realized the shortcomings of the manuscript and the necessity to revise it.AMBBEM, as an integration algorithm, an introduction to the relevant methods used is necessary. Therefore, we present the purpose and contribution of the methods used separately in the last part of Section 2.

4. The results section should be improved and compared with other recently proposed techniques and why only CT images, why do they not use MRI images also?

Our response We sincerely thank you for this suggestion and acknowledge and understand it. We have revised the experimental results according to your suggestion. We have again investigated the current state of research in this field, and we can confirm that U-net, SegNet, and Deeplabv3+ are widely used in medical image segmentation with good segmentation results. Further, according to the paper "Robust brain extraction tool for CT head images" (Neurocomputing. 2020;392: 189-195. doi:10.1016/j.neucom.2018. 12.085) and other papers, for brain extraction of head CT images, the U-net and SegNet models show good robustness, accuracy, and segmentation speed, which are superior to other existing methods. In our work, considering many model parameters such as U-net and SegNet, the segmentation speed is relatively slow, so we aim to improve the segmentation speed while ensuring good robustness and accuracy. Therefore, combined with the above situation and modification suggestions, we additionally added the SegNet model as a comparison object in the experimental results section. And in the discussion section, AMABBEM is compared theoretically with existing traditional image segmentation methods and the secondary development of medical image post-processing software to analyze various methods objectively. Why not use MRI images? It is undeniable that researchers at home and abroad have done more research on brain extraction of MRI images and proposed more methods. For the testing of MRI images and the comparison with the latest techniques, we are working on the next one. Finally, if this revision still needs to be improved, we are willing to make further changes based on the reviewers' comments.

In response to reviewer #2's five major comments and six minor comments, we made the following changes:

First of all, for the five major comments proposed:

1. The AlexNet is a classic and sound network, but not perfect today. Have you ever tried some relatively SOTA classification networks?

Our response: We are very grateful to the reviewers for this suggestion, which has broadened our thinking and improved our algorithm to some extent. Based on this suggestion, we trained and tested AlexNet, VGG19, RestNet50, and squeezeNet networks, compared and analyzed them in the experimental and discussion sections, and then selected the optimal network model for the classification task of CT images based on the comparison results.

2. The Alexnet is improved by batch normalization. How about the U-net using batch normalization?

Our response: This suggestion is complementary to the first suggestion and helps a lot to improve the manuscript. According to this suggestion, we introduce the method to improve AleNet in the algorithm design section, compare the modified version of AleNet with AleNet in the experimental section, and analyze the experimental results in the discussion section.

3. It is unfair that the AlexNet network used 5000 images for training while U-net and Deeplabv3 used 1400 images.

Our response: We thank the reviewers for this suggestion, which makes the article more rigorous and convincing. Based on this suggestion, we have redone the relevant experiments, using the same training set for six CNN models and three FCN models.

4. In experiments, the training settings, such as optimizer, learning rate, BatchSize, epochs et al., are identical for Unet and Deeplabv3+. Do they both converge and get the best results?

Our response: We sincerely thank the reviewers for this suggestion, according to which we tested different parameter settings in the training of CNN and FCN models and selected the optimal ones. In the experimental section of the manuscript, we present the training settings in detail.

5. Although the U-net network is widely used in medical image segmentation, it is not designed for segmenting CT brain images. Have you ever compared the proposed method with other methods?

Our response： I sincerely thank the reviewer for this suggestion, and I also acknowledge and understand this suggestion. We have revised the experimental results according to your suggestion. We have again investigated the current state of research in this field, and we can confirm that U-net, SegNet, and Deeplabv3+ are widely used in medical image segmentation with good segmentation results. Further, according to the paper "Robust brain extraction tool for CT head images" (Neurocomputing. 2020;392: 189-195. doi:10.1016/j.neucom.2018. 12.085) and other papers, for brain extraction of head CT images, the U-net and SegNet models show good robustness, accuracy, and segmentation speed, which are superior to other existing methods. In our work, considering a large number of model parameters such as U-net and SegNet, the segmentation speed is relatively slow, so we aim to: further improve the segmentation speed while ensuring good robustness and accuracy. Therefore, combined with the above situation and modification suggestions, we additionally added the SegNet model as a comparison object in the experimental results section. And in the discussion section, AMABBEM is compared theoretically with existing traditional image segmentation methods and the secondary development of medical image post-processing software to analyze various methods objectively. The latest technology is mostly applied to MRI images. For the expansion of the application of AMABBEM (application to MRI images) and the comparison with the latest technology, we are working on it in the next work. Finally, if this revision still does not meet the relevant requirements, we are willing to make further revisions based on the reviewers' comments.

Finally, for the 6 Minor comments:

1. Figure 6 is upside down, and "The algorithm designs a second loop as shown in Figure 7" actually means in Figure 6

Our response: We are very sorry for this error. We have replaced most of the diagrams in the manuscript, and the replacement is clearer and more concise.

2. All Figure numbers are mismatched in the text from Figure 6.

Our response： We are very sorry for such an error. We have re-edited the relevant content and checked it several times to ensure the figures correspond to the chart.

3. Lack of explanation for some settings, such as why cycle 1 is five times instead of 3 or 10 times, and Q is 0.22, and so on. Moreover, the results of Alexnet with batch normalization or without are not compared.

Our response： Thank you very much for this suggestion! In our algorithm design, q and the number of cycles are introduced in detail. The experimental part compares the modified version of Ale Net with Ale Net, and the experimental results are analyzed in the discussion section.

4. Some figures and formulas are difficult to read. Highlight the best results of each evaluation metric in tables.

Our response: We apologize that the numbers and formulas were difficult to read. We have re-edited the relevant content to make it more specific and concise. For the experimental results, we select the best results for each algorithm upon request.

5. Some sentences are hard to follow, so language and grammar editing is highly recommended.

Our response: We apologize for the sentence description issue. We have re-edited the entire article and inspected it several times to ensure grammatical correctness and clarity of language presentation.

6. PLOS authors have the option to publish the peer review history of their article (what does this mean?). If published, this will include your full peer review and any attached files.

Our response: For this suggestion, we thank the reviewers for their kind words! However, we still need to understand peer review well after reading the relevant guidelines, but we are willing to actively cooperate with the reviewers and editors if peer review helps the reviewers and editors.

Finally, we would like to express again our deep appreciation to the editors and reviewers for their valuable suggestions. We have significantly benefited from your suggestions. If the manuscript still needs to be modified, we are willing to revise it according to your requests. Finally, we wish you all the best in your review!

Yours sincerely,

Ding-Yuan Hu

---

## [Decision Letter · Decision Letter 1]

19 Sep 2023

PONE-D-23-11044R1Adaptive Mask-Based Brain Extraction Method for Head CT ImagesPLOS ONE

Dear Dr. Liang,

Thank you for submitting your manuscript to PLOS ONE. After careful consideration, we feel that it has merit but does not fully meet PLOS ONE’s publication criteria as it currently stands. Therefore, we invite you to submit a revised version of the manuscript that addresses the points raised during the review process. Please submit your revised manuscript by Nov 03 2023 11:59PM. If you will need more time than this to complete your revisions, please reply to this message or contact the journal office at plosone@plos.org. Please include the following items when submitting your revised manuscript:A rebuttal letter that responds to each point raised by the academic editor and reviewer(s). You should upload this letter as a separate file labeled 'Response to Reviewers'.A marked-up copy of your manuscript that highlights changes made to the original version. You should upload this as a separate file labeled 'Revised Manuscript with Track Changes'.An unmarked version of your revised paper without tracked changes. You should upload this as a separate file labeled 'Manuscript'.If applicable, we recommend that you deposit your laboratory protocols in protocols.io to enhance the reproducibility of your results. Protocols.io assigns your protocol its own identifier (DOI) so that it can be cited independently in the future. For instructions see: https://journals.plos.org/plosone/s/submission-guidelines#loc-laboratory-protocols. Additionally, PLOS ONE offers an option for publishing peer-reviewed Lab Protocol articles, which describe protocols hosted on protocols.io. Read more information on sharing protocols at https://plos.org/protocols?utm_medium=editorial-email&utm_source=authorletters&utm_campaign=protocols.

We look forward to receiving your revised manuscript.

Kind regards,

Khan Bahadar Khan, Ph.D

Academic Editor

PLOS ONE

Journal Requirements:

Reviewers' comments:

Reviewer's Responses to Questions

**Comments to the Author**

1. If the authors have adequately addressed your comments raised in a previous round of review and you feel that this manuscript is now acceptable for publication, you may indicate that here to bypass the “Comments to the Author” section, enter your conflict of interest statement in the “Confidential to Editor” section, and submit your "Accept" recommendation.

Reviewer #1: All comments have been addressed

Reviewer #2: All comments have been addressed

2. Is the manuscript technically sound, and do the data support the conclusions?

Reviewer #1: Yes

Reviewer #2: Yes

3. Has the statistical analysis been performed appropriately and rigorously? 

Reviewer #1: Yes

Reviewer #2: Yes

4. Have the authors made all data underlying the findings in their manuscript fully available?

Reviewer #1: No

Reviewer #2: Yes

5. Is the manuscript presented in an intelligible fashion and written in standard English?

Reviewer #1: Yes

Reviewer #2: Yes

6. Review Comments to the Author

Reviewer #1: no comments. All the comments are address. The comments have been addressed, but I was unable to locate the highlighted manuscript for tracking. I recommended this paper after reviewing the editable PDF format, as the highlighted PDF is more suitable for the reviewer

Reviewer #2: Most comments have been revised, and some additional comments are as follows.

1. AlexNet, VGG19, ResNet50, and squeezeNet mentioned in response are all classic classification networks proposed before 2016. How about the results for methods in recent years, such as EfficientNet, ResNeXt?

2. The caption of subfigure "d" in Fig 3 is missing.

3. In Figure 6, is the "Extract objects from image using area" module parallel to the main path? And what is "single-region distribution"?

4. Describe the input sizes of the first four models among the six CNN models and briefly analyze the results. Additionally, if the regular AlexNet and VGG19 also use the image with the size of 512×512×3, how would the results be?

5. The format of Mask 1 should be consistent. Some are "mask1", and others are "Mask 1". So as the "Mask2".

6. There is no caption for many figures with sub-figure.

7. In the section "Re-segmentation of the mask", what is the "region growing algorithm" for the re-segmentation of mask 2?

7. PLOS authors have the option to publish the peer review history of their article (what does this mean?). If published, this will include your full peer review and any attached files.

Reviewer #1: No

Reviewer #2: No

---

## [Author Response · Author response to Decision Letter 1]

30 Sep 2023

Dear Editor：

I am very grateful to the editor for replying to my busy schedule, and I would like to thank the reviewers again for their valuable comments. In publishing a paper, receiving comments and guidance from the reviewers is crucial for better solving practical problems. We are lucky because the reviewers' comments are professional and rigorous, which opened our eyes and standardized the manuscript. We answered each reviewer's comments and adjusted the manuscript and algorithms. Our specific answers and revisions are listed below:

First of all, we carefully reviewed the references for the journal's requirements for a reference list, combined with the actual situation of manuscript changes. We made the following adjustments:

1.To further highlight the characteristics of CT imaging and to facilitate the reader's understanding of the latter. We replaced the original reference ' Parameter Calibration and Image Reconstruction of CT System ' with ' Improving Sensitivity on Identification and Delineation of Intracranial Hemorrhage Lesion Using Cascaded Deep Learning Models '.

2.Because the manuscript excludes the use of AlexNet, we also exclude the reference to the literature "ImageNet Classification with Deep Convolutional Neural Networks."

3.The manuscript also tested the EfficientNet network and used the SE and CBAM modules. I have added three articles: ' EfficientNet: Rethinking Model Scaling for Convolutional Neural Networks, '' CBAM: Convolutional Block Attention Module ' and ' Squeeze-and-Excitation Networks '.

4.To make it easier for readers to understand the characteristics of the region growing algorithm, we also cite the literature "A review on brain tumor segmentation of MRI images."

Second, we responded to reviewer #1's comment as follows:

1. no comments. All the comments are address. The comments have been addressed, but I was unable to locate the highlighted manuscript for tracking. I recommended this paper after reviewing the editable PDF format, as the highlighted PDF is more suitable for the reviewer

Our response: We are very grateful to the reviewer for this comment and for recognizing the last revision. Further, we apologize for the reviewer's inability to locate the highlighted manuscript and thus follow it up; this time, we have made the revision marks on the original manuscript much clearer.

Third, we have made the following responses and revisions to the six comments made by reviewer 2:

1. AlexNet, VGG19, ResNet50, and squeezeNet mentioned in response are all classic classification networks proposed before 2016. How about the results for methods in recent years, such as EfficientNet, ResNeXt?

Our response: Thank you very much to the reviewer for this comment! This comment once again broadens our horizons, and introducing more excellent networks further improves the superiority of our algorithm. As the reviewer said, AlexNet, VGG19, ResNet50, and squeezeNet are indeed relatively old, and the subsequent more excellent EfficientNet and ResNeXt are more worth trying. Therefore, we also tested the EfficientNet network and again delved into the ResNet family of networks. We found that ResNeXt was obtained by improving ResNet while also noting that SE_ResNet50... and other networks. Finally, we try to improve on ResNet50 by adding a CBAM module to the Stem Block session of ResNet50 and then adding an SE module to each of the four Bottleneck sessions. Then, we fine-tune the training set and leave the test set 1 unchanged. Finally, VGG19, ResNet50, squeezeNet, EfficientNet, and the improved ResNet50 are trained on the training set; during the training process, multiple sets of optimizers and hyperparameters are tested for each network model, and the best optimizer and hyperparameters are selected. The experimental results show that the improved ResNet50 gets a significant improvement in both Accuracy and AP values.

In addition, we again performed further inspection and correction of test set 2 with the assistance of a radiologist, but of course, the adjustments were minimal. Finally, each FCN and AMBBEM were tested again. The results showed that the segmentation speed of AMBBEM was further improved after the replacement of the CNN, which was attributed to the improved resnet50 with higher accuracy and a lower number of parameters than the previous CNN, which resulted in a faster classification speed!

Of course, the article has been adjusted accordingly in the Abstract, Algorithm Design, Experimental Results, and Talking About sections because of the adjustments to the CNN network.

2. The caption of subfigure "d" in Fig 3 is missing.

Our response: We apologize for this comment! We have corrected this error.

3. In Figure 6, is the "Extract objects from an image using area" module parallel to the main path? And what is "single-region distribution"?

Our response: We thank the reviewer for this comment! This evaluation made us aware of errors and deficiencies in Figure 6 and its presentation. We have revised Figure 6. In addition, the term "single-region distribution" indicates that the brain tissue is a single block in a head CT image, as opposed to the brain being distributed in multiple blocks. Finally, we have made adjustments in the presentation of the manuscript to address these two issues to facilitate better understanding by the readers.

4. Describe the input sizes of the first four models among the six CNN models and briefly analyze the results. Additionally, if the regular AlexNet and VGG19 also use the image with the size of 512×512×3, how would the results be?

Our response: Much respect to the reviewers for their dedication and seriousness! We should describe the CNN networks' input sizes in further detail. In addition, we also tested the normal AlexNet and VGG19 networks with the input size of 512×512×3, respectively. The results show that, due to the setting of the three fully connected layers in the VGG19, the input layer size is fixed (224×224×3), and the input size can not be changed. For the normal AlexNet, the adjustment of the input size does not have any significant effect on it. To show the experimental conclusions more concisely. We only offer the test results of VGG19, ResNet50, squeezeNet, EfficientNet, and the improved ResNet50 and exclude the results of ordinary AlexNet and improved AlexNet from the manuscript content.

5. The format of Mask 1 should be consistent. Some are "mask1", and others are "Mask 1". So as the "Mask2".

Our response: I thank the reviewer for this comment and apologize for this oversight; we have standardized the formatting involved.

6. There is no caption for many figures with sub-figures.

Our response: We have been discussing this evaluation carefully and have always found the reviewer's suggestion more reasonable. We have added a description of the images with subplots based on the format of the latest published paper in the journal.

7. In the section "Re-segmentation of the mask," what is the "region growing algorithm" for the re-segmentation of mask 2?

Our response: Thanks to the reviewers for this comment! We apologize for the lack of clarity here. Region growing algorithm is an image segmentation technique. The basic idea will be to combine pixels with similar criteria to form regions based on certain discriminative criteria. The main step is to find a seed pixel for each region to be segmented as a starting point for growth, and then according to a certain discriminative criterion, the similar pixels around the seed pixel are discriminated, and the pixels with higher similarity are merged so that they germinate and grow like a seed. We have adapted the presentation part of the region growing algorithm in the article, in addition to adding references to facilitate the reader's understanding.

In addition, we have negotiated and hope to make the following adjustments to the paper:

1.The paper was published with equally significant contributions from Shiya Qu, so we would like to list Shiya Qu as the 1st set of equal contributors. In addition, we would like to add Qingyan Zhang (radiologist) as the sixth author, considering that he has provided a lot of help in the process of publication and revision of the paper. Of course, this change must be made with the editor's permission. We are willing to follow the editor's arrangement if the editor feels something inappropriate, and the editor's arrangement will prevail.

Finally, we checked and proofread the article several times and adjusted some words and grammar. For the supporting material, we were keen to follow the request to share experimental materials. However, some of the datasets were too large, and multiple uploads failed. For this reason, we can only split the training set (S1) into four parts for uploading. The file "S1_CNN_Training set" contains the dataset required for CNN training, which has already been categorized and can be used directly. File "S1_FCN_Training set_Label" contains the labels required for FCN model training, "S1_FCN_Training set_Set_1" and "S1_FCN_Training set_Set_2" files together contain 1400 images required for FCN model training, which need to be combined into one piece after downloading. Files S2 through S5 are as described in the manuscript. Where S5 contains instructions for use and an improved ResNet50 network model in addition to the code. Also, we mention in the manuscript that readers can also direct questions about the experimental material to the corresponding author.

---

## [Decision Letter · Decision Letter 2]

23 Nov 2023

Adaptive Mask-Based Brain Extraction Method for Head CT Images

PONE-D-23-11044R2

Dear Dr. Liang,

We’re pleased to inform you that your manuscript has been judged scientifically suitable for publication and will be formally accepted for publication once it meets all outstanding technical requirements.

Kind regards,

Khan Bahadar Khan, Ph.D

Academic Editor

PLOS ONE

Additional Editor Comments (optional):

Reviewers' comments:

Reviewer's Responses to Questions

**Comments to the Author**

1. If the authors have adequately addressed your comments raised in a previous round of review and you feel that this manuscript is now acceptable for publication, you may indicate that here to bypass the “Comments to the Author” section, enter your conflict of interest statement in the “Confidential to Editor” section, and submit your "Accept" recommendation.

Reviewer #2: All comments have been addressed

Reviewer #3: (No Response)

2. Is the manuscript technically sound, and do the data support the conclusions?

Reviewer #2: Yes

Reviewer #3: Yes

3. Has the statistical analysis been performed appropriately and rigorously? 

Reviewer #2: Yes

Reviewer #3: Yes

4. Have the authors made all data underlying the findings in their manuscript fully available?

Reviewer #2: Yes

Reviewer #3: Yes

5. Is the manuscript presented in an intelligible fashion and written in standard English?

Reviewer #2: Yes

Reviewer #3: Yes

6. Review Comments to the Author

Reviewer #2: The authors have addressed all comments raised in a previous round and this manuscript is now acceptable for publication. No further comment.

Reviewer #3: 1-Consider restructuring the content to present contributions in a concise and organized manner, utilizing bullet points for clarity. This would help both the authors and readers to quickly grasp the significant contributions made in the study.

2- The manuscript exhibits numerous formatting and grammatical errors, including inconsistencies in headings.

3- The conclusion section needs revision to ensure a clear distinction between conclusions and methodologies. Please rewrite the conclusion section, focusing solely on summarizing the key findings and drawing conclusions without delving into the details of the methodologies employed.

4- The manuscript would benefit from a more comprehensive discussion of the technical aspects of previously published work. It is essential to delve into the details of the relevant literature, providing a thorough analysis of the technical methodologies and approaches employed in previous studies. Please consider expanding the discussion on the technical aspects of previously published work in the revision.

5- The overall quality of the paper is commendable. The authors have done a good job in presenting their work. However, it would be beneficial to address specific points mentioned in the earlier comments for further improvement. Keep up the good work!

7. PLOS authors have the option to publish the peer review history of their article (what does this mean?). If published, this will include your full peer review and any attached files.

Reviewer #2: No

Reviewer #3: **Yes: **Tehreem Awan

---

## [Editor Report · Acceptance letter]

12 Dec 2023

PONE-D-23-11044R2 

Adaptive Mask-Based Brain Extraction Method for Head CT Images 

Dear Dr. Liang:

I'm pleased to inform you that your manuscript has been deemed suitable for publication in PLOS ONE. Congratulations! Your manuscript is now with our production department. 

Kind regards, 

on behalf of

Dr. Khan Bahadar Khan 

Academic Editor

PLOS ONE